
# Satellite drag effects due to uplifted oxygen neutrals during
# super magnetic storms
Gurbax S. Lakhina[1] and Bruce T. Tsurutani[2]
[1]Indian Institute of Geomagnetism, New Panvel (W), Navi Mumbai, India.
[2]Jet Propulsion Laboratory, California Institute of Technology, Pasadena, CA, USA.
*Correspondence to*: Gurbax S. Lakhina (gslakhina@gmail.com)
**Abstract:** During intense magnetic storms, prompt penetration electric fields (PPEFs) through
$\mathbf{E} \times \mathbf{B}$ forces near the magnetic equator uplift the dayside ionosphere. This effect has been called
the "dayside superfountain effect". Ion-neutral drag forces between the upward moving
$O^+$(oxygen ions) and oxygen neutrals will elevate the oxygen atoms to higher altitudes. This
paper gives a linear calculation indicating how serious the effect may be during an 1859-type
(Carrington) superstorm. It is concluded that the oxygen neutral densities produced at low-
Earth-orbiting (LEO) satellite altitudes may be sufficiently high to present severe satellite drag. It
is estimated that with a prompt penetrating electric field of ~20 mV/m turned on for 20 min, the
O atoms and $O^+$ ions are uplifted to 850 km where they produce about 40 times more satellite
drag per unit mass than normal. Stronger electric fields will presumably lead to greater uplifted
mass.

**1. Introduction**
Prompt penetration of interplanetary electric fields (IEFs) to the dayside equatorial ionosphere
has been known for a long time (Obayashi 1967; Nishida 1968; Kelley et al. 1979). It has been
shown that during super magnetic storms, defined as storms with Dst < -250 nT (Tsurutani et al.
1992; Echer et al. 2008), the prompt penetrating electric fields (PPEFs) associated with large IEF
intervals can last for more than several hours in the ionosphere (Maruyama et al. 2004; Mannucci
et al. 2005; Sahai et al. 2005; Huang et al. 2005). Intense dawn-to-dusk (eastward viewing from
the northern hemisphere) PPEFs uplift the dayside plasma to higher altitudes and magnetic
latitudes due to $\mathbf{E} \times \mathbf{B}$ drifts (Tsurutani et al. 2004; Mannucci et al. 2005; Verkhoglyadova et al.
2007). The ionospheric electron-ion recombination rate is much slower at higher altitudes





(Tsurutani et al.   2005), thus the "old" uplifted ionosphere is more or less stable. Solar
photoionization replaces the displaced plasma at lower altitudes, increasing the total electron
content (TEC) of the ionosphere. After the PPEF subsides, the   plasma flows down along the
magnetic field lines to even greater magnetic latitudes. This overall process is named as the
"dayside superfountain effect".

During superstorms, the vertical TEC values are found to increase to several times quiet time
values across the dayside ionosphere at low and middle latitudes. This has been empirically
observed by satellite and from ground-based GPS receivers. Apart from the dayside
superfountain effect, which occurs during the first few hours of a superstorm, there is another
mechanism called the "disturbance dynamo" (Blanc & Richmond 1980; Fuller-Rowell et al.
1997). The latter is caused by particle precipitation and atmosphere heating in the auroral zone
during the superstorms and consequential equatorward-directed neutral winds due to this heating
process. However, all superstorms are not alike as they have different peak intensities and
associated convection electric fields (Gonzalez et al. 1994). Therefore, the PPEFs produce
different effects in terms of TEC enhancements, poleward-shifting of equatorial ionization
anomaly (EIA) peaks (Namba & Maeda 1939; Appleton 1946) from the typical quiet time
positions at ~ $\pm10°$, compositional changes, etc. Studies of the Bastille day (15 July 2000)
superstorm (Basu et al. 2001, 2007;  Kil et al. 2003, Yin et al. 2004; Rishbeth et al. 2010), the
Halloween (30 October 2003) superstorm (Tsurutani et al. 2004, 2007, 2008; Mannucci et al.
2005; Verkhoglyadova et al. 2007), and some other superstorms events (Foster et al. 2004; Lin et
al. 2005; Immel et al. 2005; Mannucci et al. 2008, 2009) clearly illustrate the above point.


Using a modified version of the low- to mid-latitude ionosphere code SAMI2 (Sami2 is Another
Model of the Ionosphere) of the Naval Research Laboratory (NRL) (Huba et al. 2000, 2002),
Tsurutani et al. (2007) have studied the $O^+$ ion uplift in the dayside ionosphere due to first ~ 2
hours of  PPEFs during the  30 October 2003 (Halloween) superstorm. Their simulations clearly
show the dayside $O^+$ ions uplifted to higher altitudes (~600 km) and higher magnetic latitudes
(**MLAT**) (~ $\pm25°$ to 30°), forming highly displaced EIA peaks. The rapid upward drift of the $O^+$
ions causes neutral oxygen (O) uplift due to ion-neutral drag forces. They also find that above ~



400 km altitude, the neutral oxygen atom densities within the displaced EIAs increase
substantially over their quiet time values.

Recently, Tsurutani et al. (2012) have modeled the 1-2 September 1859 Carrington storm using
the modified SAMI2 code (Verkhoglyadova, 2007, 2008).  This superstorm's intensity was the
highest in recorded history, Dst ~ -1760 nT (Tsurutani et al. 2003; Lakhina et al. 2012).  The
storm-time electric field has been estimated to have been ~20 mV/m. Similar features to the 30
October 2003 storm were found, but all effects were more severe.  The EIAs were found to be
located at ~ 500 to 900 km altitude with broad peaks located at ~ ±25° to 40° MLAT. In this
paper, we study the uplift of neutral oxygen O atoms due to the ion-neutral drag force during an
1859-type superstorm. The possible satellite drag effects on Low Earth Orbiting (LEO) satellites
will be discussed.

## 73  2. Change in neutral O atom densities  due to ion-neutral drag

When O$^+$ ions drift rapidly upwards through the neutral atmosphere (under the influence of an
**E**×**B** force associated with the PPEFs during an 1859-type superstorm), they exert an ion-neutral
drag force on the neutral atoms and will uplift them (Tsurutani et al., 2007). A simplified ion-
neutral momentum exchange is given by (Baron & Wand 1983; Kosch et al. 2001):
$$\frac{\partial U}{\partial t} = \frac{1}{\tau_{in}}(V_d - U) \tag{1}$$

where $U$ is the vertical speed of the neutral oxygen atom due to ion-neutral drag force,  $V_d =$
**E**×**B**/B$^2$ is the O$^+$  vertical drift  due to the **E**×**B** force, and  $\tau_{in}$ is the ion-neutral coupling time
constant (Killeen et al. 1984) given by:
$$\tau_{in} = \frac{n_0}{n_i\, \nu_{in}} \tag{2}$$

In Eq. (2), $n_o$ is the neutral oxygen O atom density, $n_i$ is the O$^+$ ion density, and $\nu_{in}$ is the ion-
neutral collision frequency.



Following Tsurutani et al. (2007), we calculate the ion-neutral coupling time, $\tau_{in}$, for a
representative altitude of ~ 340 km. We obtain $\nu_{in}$ from the expression given by Bailey & Balan
(1996), i.e., $\nu_{in} = 4.45 \times 10^{-11} . n_o .T^{1/2} .(1.04 - 0.067 \log_{10} T)^2$, where T is average of the O and O$^+$
temperatures. Using O and O$^+$ temperatures of ~$10^3$ K and noon-time densities of $n_o = 1.1 \times 10^9$
cm$^{-3}$ and $n_i = 3.5 \times 10^6$ cm$^{-3}$, we get $\tau_{in}$ ~ 5 min.
Considering the initial (boundary) conditions at the reference altitude (z ~ 340 km) as $U = 0$ at t
= 0, the solution to Eq.(1) can be written as
$$U = V_d\left[1 - \exp\left(-\frac{t}{\tau_{in}}\right)\right] \qquad (3)$$

The uplift of the neutrals will cause changes in their density with altitude, z. To first order, the
continuity equation for O can be written as:

$$\frac{\partial n}{\partial t} + U\frac{\partial n_o}{\partial z} = 0 \qquad (4)$$


On substituting U from Eq.(3) and integrating Eq.(4) with the boundary condition that $n = n_o$ at t
= 0, we get the change in neutral density as:
$$\delta n = \frac{V_d}{H}\left[t - \tau_{in}\left(1 - \exp\left(-\frac{t}{\tau_{in}}\right)\right)\right] \qquad (5)$$


where $\delta n = (n - n_o)/n_o$, and $H = \left(\frac{1}{n_o}\frac{\partial n_o}{\partial z}\right)^{-1}$ is the oxygen neutral scale height. Equation (5) implies
that neutral density time dependence at progressively higher altitude layers will be affected  by
the arrival of neutrals uplifted from below due to the ion-neutral drag.  We must emphasize that
Eq.(5) gives only the first-order estimates. Implicitly it is assumed that the pressure gradient and
gravity effects balance each other during the uplift. For   more accurate estimates, one has to
consider the nonlinear coupling terms along with the inclusion of gravity, pressure gradients,



viscosity and the effects of heating and expansion during the uplift process. The decrease of the
ambient oxygen densities are also not taken into account in the above estimate.  All of these will
have to be considered in a fully self consistent nonlinear code.

**3.  Satellite drag due to $O^+$ and O enhanced fluxes during superstorms**
Drag force per unit mass on a satellite moving through the Earth's atmosphere is given by
(Chopra 1961; Gaposchkin & Coster 1988; Moe & Moe 2005; Pardini et al. 2010; Li 2011):
$$F = \tfrac{1}{2}C_D(\tfrac{A}{M})(mn)V_s^2 + \tfrac{1}{2}C_{Di}(\tfrac{A}{M})(m_in)V_s^2 \qquad\qquad (6)$$
where $C_D$ represents the neutral drag coefficient due to impingement of O atoms on the satellite
surface and $C_{Di}$ is the ion (Coulomb) drag coefficient due to scattering of $O^+$ ions by the  satellite
potential (a satellite moving in an ionized atmosphere acquires an electric charge mainly through
collisions with charged particles), $m$ is the mass of neutral atom (O), and $m_i$ is the mass of  $O^+$
ions, $A$ is the satellite cross-section area perpendicular to the velocity vector, $M$ the mass of the
satellite, and $V_s$ is the satellite velocity with respect to the atmosphere. The drag coefficients $C_D$
and $C_{Di}$ have been discussed theoretically and calculated from empirical observations of satellite
deceleration and other data by many workers (Chopra 1961; Cook 1965, 1966; Fournier 1970;
Gaposchkin & Coster 1988; Moe & Moe 2005; Moe et al. 1998; Pardini et al. 2010; Li 2011).
The information on the gas-surface interaction on the surface of the satellite and contamination
of the satellite surface due to the adsorbed atomic oxygen, are essential to accurately determine
the drag coefficients (Chopra 1961; Cook 1965, 1966; Pardini et al. 2010). Various studies
(Chopra 1961; Cook 1965, 1966; Pardini et al. 2010 ) show that for spherically- or cylindrically-
shaped satellites, the neutral drag coefficient $C_D$ varies from ~ 2.0 to 2.8 between altitudes of z =
~ 200 to 800 km with a most commonly used value of  $C_D = 2$. In contrast the Coulomb drag
coefficient $C_{Di}$ varies widely with altitude, e.g., $C_{Di} = 7\times10^{-5}$, 0.32 and 6.1 at z = 250, 500 and
800 km, respectively (Chopra, 1961; Li, 2011). The area to mass ratio of the satellite, $A/M$,  can
have values of ~ 0.038 - 0.285 cm$^2$/g, which obviously varies from satellite to satellite. A typical
value of a satellite payload is $A/M = 0.1$ cm$^2$/g (Gaposchkin & Coster 1988; Pardini et al. 2010),
which we will use in our following calculations. A typical value for the LEO satellite speed with



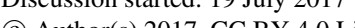
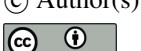

respect to the atmosphere is $V_s \sim 7.5$ km s$^{-1}$. Thus our calculations of $n_o$ (O densities) during
superstorms can be used to calculate the drag force on LEO satellites by using Eq. (5).

In Table 1, for the super magnetic storm of 1-2 September 1859, we have given the estimates of
altitude, z, reached by uplifted O atoms from the integration of Eq. (3) (column 2), change in O
density from Eq. (5), $\delta n$ (column 3), the Coulomb drag coefficient, $C_{Di}$ extrapolated from the
values given by Chopra (1961) and Li (2011) (column 4), and the drag force per unit mass, $F$
calculated from Eq. (6) (column 5) for various values of time, t, after the application of 20 mV/m
PPEF in the equatorial ionosphere (with a constant $B_0 = 0.35\times10^{-4}$ T) (column 1). A constant
neutral drag coefficient $C_D = 2.0$, $V_s = 7.5$ km s$^{-1}$, and $A/M = 0.1$ cm$^2$/g and for a scale height of
$H \sim 50$ km are assumed. The reference altitude is taken at z = 340 km with the oxygen atom (O)
mass density $mn_o = 2.94361\times10^{-14}$ g cm$^{-3}$. The background neutral O density is assumed to
decrease exponentially with altitude with a scale height of $H = 50$ km. It is interesting to note
that in just 20 minutes (t = 1200 s), the uplifted O atoms reach an altitude of z = 856 km from the
reference  altitude of 340 km at t = 0. The drag force per unit mass on the satellite is $F = 0.0017$
cm s$^{-2}$ at z = 340 km (t = 0) and it increases to $F = 0.0692$ cm s$^{-2}$ at an altitude of z = 856 km (t =
1200 s), an increase of more than 40 times! As one can see from Column 4, the Coulomb drag
coefficient increases with altitude, therefore its contribution to $F$ increases as O$^+$ ions and O
neutral atoms are uplifted by the action of PPEFs via the $\mathbf{E}\times\mathbf{B}$ force. We find that Coulomb drag
dominates over the  neutral drag at altitudes above $\sim 750$ km.
**4. Conclusions**
We have done preliminary estimates of the drag force per unit mass on typical low Earth orbiting
satellites moving through the ionosphere during super magnetic storms, like the Carrington 1-2
September 1859 event. A simple first-order model is employed to calculate the changes in
density of the neutral O atoms at different altitudes due to ion-neutral drag between the uplifted
O$^+$ ions and O neutral atoms. The uplifted O$^+$ ion speeds result from the $\mathbf{E}\times\mathbf{B}$ force from the
PPEFs. It should be noted that there is no expansion of the column of gas from  340 km  to 850
km, rather the entire column of atmosphere is uplifted by the EXB  force. There may be a slight
increase in the temperature of the O atoms due to the friction with O$^+$ ions. Consequently, as the



pressure  remains the same or is slightly increased, it will more or less balance the gravity during
the first 20 minutes or so. Eventually, the nonlinear coupling, gravity, pressure gradients and
viscosity effects will dominate and will stop the uplift of the neutrals. Therefore, this simple
model may be reasonable for the first ~ 20 minutes after the onset of the PPEF in the ionosphere.
In this paper, we have not considered the effect of Joule heating in increasing the neutral
densities and temperatures during super magnetic storms.  The Joule heating occurs  in the
auroral regions, but it may come down to lower latitudes during superstorms. Therefore, the
effects due to Joule heating are important primarily at high latitudes initially, and  such increases
are expected to manifest at equatorial latitudes after 2-3 hours or more.  However our mechanism
will occur near the equator and at middle latitudes.  At middle latitudes the two mechanisms
would most likely merge.  Furthermore,  it can be estimated that an increase in neutrals
temperature of 200 K, say from 800 to 1000 K,  will cause a factor of 10 increase in the O
density at 850 km just by the scale height effect.  A 400 degree increase in temperature would
increase the O density by a factor of 100 at 850 km. This is the same increase that is obtained
from our  proposed mechanism. It should be remembered that the process of uplift of neutral O
atoms due to penetration electric fields during super magnetic storms occurs during the span of
20 minutes. In reality, an increase in the temperature of neutrals by 200 K to 400 K in 20 minutes
at equatorial latitudes by the Joule heating or any other process during magnetic storms has not
been observed.

It is shown that in just ~ 20 minutes after the action of a 20 mV/m PPEF in the equatorial
ionosphere, the neutral O atoms (and also $O^+$ ions) are uplifted to an altitude of $z = 856$ km from
a reference level of $z = 340$ km. A typical spherically- or cylindrically- shaped satellite moving
through the ionosphere at altitudes of ~ 850 km would experience a  40 times more drag per unit
mass than normal. If larger IEFs associated with either superflares (see Maehara et al.  2012,
Tsurutani and Lakhina, 2014)  or during extreme magnetic storms stronger than the Carrington
storm (Vasyliunas 2011, Lakhina and Tsurutani, 2016), are  imposed on the magnetosphere then
larger scale PPEFs will be imposed on the dayside ionosphere with even greater O atom uplift.
We do not know when such cases can occur at the Earth, but we cannot exclude the possibility
at this time.




Acknowledgements: Portion of this work was done at NASA Jet Propulsion Laboratory,
California Institute of Technology, Pasadena, CA, USA. GSL thanks the Indian National Science
Academy for support under the Senior Scientist Scheme.

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

Table 1: Estimates of the altitude, z, attained by uplifted O atoms, relative change in O density,
$\delta n$, Coulomb drag coefficient (extrapolated from Chopra (1961) and Li (2011)), $C_{Di}$, drag force
per unit mass on satellite, $F$, at different times, t, after the onset of a PPEF of 20 mV/m in a
Carrington (1-2 September 1859) super magnetic storm. The reference altitude is taken at z =
340 km with the Earth's magnetic field, $B_0 = 0.35\times10^{-4}$ T, and the oxygen atom (O) mass density
$mn_o = 2.94361\times10^{-14}$ g cm$^{-3}$. The background neutral O density is assumed to decrease
exponentially with altitude with a scale height of $H = 50$ km. The other parameters are: neutral
drag coefficient $C_D = 2.0$, $V_s = 7.5$ km s$^{-1}$, and area to mass ratio of satellite, $A/M = 0.1$ cm$^2$/g,
and the scale height, $H = 50$ km.

| t = time after onset of PPEFs, in s | z = altitude attained, in km | $\delta n$ = change in O density | $C_{Di}$ = Coulomb drag coefficient | $F$= drag force per unit mass, in cm s$^{-2}$ |
|---|---|---|---|---|
| 0 | 340 | 0 | 0.01 | 0.0017 |
| 300 | 402 | 1.26 | 0.15 | 0.0028 |
| 600 | 534 | 3.88 | 0.32 | 0.0075 |
| 900 | 690 | 7.01 | 1.43 | 0.0199 |
| 1200 | 856 | 10.32 | 6.1 | 0.0692 |
