# Peer review of "Satellite drag effects due to uplifted oxygen neutrals during super magnetic storms"

_Nonlinear Processes in Geophysics, 2017_

## Referee Comment (RC1) · Anonymous Referee #1 · 21 Aug 2017

I have examined the manuscript npg-2017-33 submitted by Gurbax S. Lakhina and Bruce T. Tsurutani entitled "Satellite drag effects due to uplifted oxygen neutrals during super magnetic storms". This paper identifies basic aspects of plasma behavior that have not been previously considered in calculations of satellite drag. The work is creative, based on solid principles, and makes definitive quantitative predictions of relevance to space scientists. I recommend the paper for publication in its present form.

---

## Referee Comment (RC2) · Y. Narita (Referee) · 8 Sep 2017

The manuscript develops a simplified model for the oxygen uplift from the low-altitude ionosphere to the higher altitude caused by the enhanced E x B drift effect during the extreme or major geomagnetic storm time, and applies the oxygen fluxes to predicting the satellite drag by taking the Carrington super-magnetic storm event as an example.

The manuscript is a beautiful application involving the space science (the Sun-Earth relation), the physics of the ionosphere, and the engineering aspect (satellite drag estimate). The model for the oxygen uplift (section 2) is rather simple, but nevertheless contains the essence of the physical process (uplift flow estimate, drag force between

plasma and neutral, scale height, and continuity). The model is developed for the linear treatment of the uplift, but the authors address what effects need to be considered when upgrading into the nonlinear treatment.

The authors apply the oxygen density profile (from section 2) to the model of the satellite drag force (Equation 6, section 3), and find that the drag force can significantly vary from a lower to a higher altitude by a factor of about 40. The authors also find that the electrostatic drag force (Coulomb effect) dominates over that of the neutral gas at higher altitudes above 750 km.

The manuscript reads well. The logic and the calculations are easy to follow. And the study is concise with a clear message to the audience. The manuscript will also serve as a beautiful example of writing a paper for the young students. I enjoyed reading the manuscript. I have only minor comments in a hope of improving the quality of the manuscript a bit (the authors may disagree). In any case, I recommend the manuscript for a prompt publication.

page 2, line 37. "GPS" appears for the first time in the main text. I propose to rewrite into "GPS (Global Positioning System)" such that the readers can continue reading the paper without being disturbed by the acronym.

page 4, line 91. I wonder how the reference altitude (340 km) was chosen. Can the authors say if it is conventional or maybe if it is from a computational reason?

page 4, line 107 to page 5, line 109. Should the advection of the O-atom flow (U dot nabla U) be included for the nonlinear treatment, too? Turbulence physicists might find the advection term as interesting as the other effects.

page 5, line 127. "adsorbed" should read "absorbed".

page 6, line 140. As a reader, I prefer to see "we give the estimates of..." rather than "we have given the estimates..." because the discussion sounds on-going. But the authors can decide.

page 6, line 164. It is better to write "EXB" as \mathbf{E} \times \mathbf{B} coherently in the text.

---

## Referee Comment (RC3) · Anonymous Referee #3 · 14 Sep 2017

This paper addressed on uplifted oxygen neutrals due to the prompt penetrating electric fields in the dayside ionosphere, and discussed the drag force on a low Earth orbiting satellite during super magnetic storms, like the Carrington superstorm. The physical process of the uplifted oxygen ions and atoms was concisely documented, and the satellite drag force was reasonably estimated. However, I wonder if the ionospheric atmosphere in the night side may be depressed during the penetration of electric field, and then the drag force may be reduced in the night side. Therefore, the drag force averaging one orbiting cycle may be compensated in some sense.

[Figure]

2017-33, 2017.

---

## Author Comment (AC1) · 30 Oct 2017

**Response to comments by the Referees on the manuscript npg-2017-33 entitled,"Satellite drag effects dueto uplifted oxygen neutrals during super magneticstorms" by Gurbax S. Lakhina and Bruce T.Tsurutani**

We thank the Referees for carefully reading the manuscript, and for their very useful comments. We have revised the paper in light of their comments. In our point-to-point Reply below, the comments by the Referees are shown in regular fonts, and our reply in *Italic* fonts.

**Anonymous Referee #1 (RC1)**
Comment:
I have examined the manuscript npg-2017-33 submitted by Gurbax S. Lakhina and Bruce T. Tsurutani entitled "Satellite drag effects due to uplifted oxygen neutrals during super magnetic storms". This paper identifies basic aspects of plasma behavior that have not been previously considered in calculations of satellite drag. The work is creative, based on solid principles, and makes definitive quantitative predictions of relevance to space scientists. I recommend the paper for publication in its present form.

*Reply: Thank you very much for your encouraging comments and recommending the paper for publication.*

**Y. Narita (Referee#2)(RC2)**

Comment:

The manuscript develops a simplified model for the oxygen uplift from the low-altitude ionosphere to the higher altitude caused by the enhanced E x B drift effect during the extreme or major geomagnetic storm time, and applies the oxygen fluxes to predicting the satellite drag by taking the Carrington super-magnetic storm event as an example. The manuscript is a beautiful application involving the space science (the Sun-Earth relation), the physics of the ionosphere, and the engineering aspect (satellite drag estimate). The model for the oxygen uplift (section 2) is rather simple, but nevertheless contains the essence of the physical process (uplift flow estimate, drag force between plasma and neutral, scale height, and continuity). The model is developed for the linear treatment of the uplift, but the authors address what effects need to be considered when upgrading into the nonlinear treatment.
The authors apply the oxygen density profile (from section 2) to the model of the satellite drag force (Equation 6, section 3), and find that the drag force can significantly vary from a lower to a higher altitude by a factor of about 40. The authors also find that the electrostatic drag force (Coulomb effect) dominates over that of the neutral gas at higher altitudes above 750 km.
The manuscript reads well. The logic and the calculations are easy to follow. And the study is concise with a clear message to the audience. The manuscript will also serve as a beautiful example of writing a paper for the young students. I enjoyed reading the manuscript. I have only minor comments in a hope of improving the quality of the manuscript a bit (the authors may disagree). In any case, I recommend the manuscript for a prompt publication.

*Reply: We thank the referee for going through the manuscript critically and recommending it for publications. We have taken all your suggestions into account in the revised manuscript.*

Comment:
page 2, line 37. "GPS" appears for the first time in the main text. I propose to rewrite into "GPS (Global Positioning System)" such that the readers can continue reading the paper without being disturbed by the acronym.

*Reply: Done. Thank you.*

Comment:
page 4, line 91. I wonder how the reference altitude (340 km) was chosen. Can the authors say if it is conventional or maybe if it is from a computational reason?

*Reply:Thank you for raising this issue. For the calculations, the reference level at 340 km was chosen because it is near the equatorial ionization anomaly (EIA) density peak location where the ion-neutral drag is expected to be approximately a maximum (Tsurutani et. al., 2007). This is included in the text now.*

Comment:
page 4, line 107 to page 5, line 109. Should the advection of the O-atom flow (U dot nabla U) be included for the nonlinear treatment, too? Turbulence physicists might find the advection term as interesting as the other effects.

*Reply:Good suggestion. Done. Thank you.*

Comment:
page 5, line 127. "adsorbed" should read "absorbed".

*Reply: done. Thank you.*

Comment:
page 6, line 140. As a reader, I prefer to see "we give the estimates of..." rather than "we have given the estimates..." because the discussion sounds on-going. But the authors can decide.

*Reply: Done. Thank you.*

Comment:
page 6, line 164. It is better to write "EXB" as nmathbf{E} ntimes nmathbf{B} coherently in the text.

*Reply: Done. Thank you.*

**Anonymous Referee #3 (RC3)**

This paper addressed on uplifted oxygen neutrals due to the prompt penetrating electric fields in the dayside ionosphere, and discussed the drag force on a low Earth orbiting satellite during super magnetic storms, like the Carrington superstorm. The physical process of the uplifted oxygen ions and atoms was concisely documented, and the satellite drag force was reasonably estimated.

*Reply: We thank the referee for going through the manuscript critically and for the encouraging comments.*

Comment:
However, I wonder if the ionosphericatmosphere in the night side may be depressed during the penetration of electric field,and then the drag force may be reduced in the night side. Therefore, the drag force averaging one orbiting cycle may be compensated in some sense.

*Reply: Yes, we agree that the nightside ionosphere will be depressed due to change of sign of $E \times B$ drift (i.e., downward drift instead of uplift).This was mentioned in the Tsurutani et al. 2004 discovery paper. However, as the neutral O atom density will increase sharply at lower altitudes, the relative change in O atom density due to ion-neutral drag force would be relatively small, and that too at altitudes lower than the reference level. Since at the higher altitudes, the neutral O density is expected to remain more or less unchanged, the satellite on the nightside will not feel any extra drag force due to $E \times B$ drift. Therefore, we do not expect that the nightside ionosphere can compensate for the extra dayside satellite drag due to uplifted O atom over a satellite orbit. This point is addressed by adding an extra text on page 7 in the revised manuscript.*